# Effects of Thermal Annealing on Optical and Microscopic Ferromagnetic Properties in InZnP:Ag Nano-Rods

**DOI:** 10.3390/nano12234200

**Published:** 2022-11-25

**Authors:** Juwon Lee, Yoon Shon, Younghae Kwon, Ji-Hoon Kyhm, Deuk Young Kim, Joon Hyun Kang, Chang-Soo Park, Kyoung Su Lee, Eun Kyu Kim

**Affiliations:** 1Quantum-Functional Semiconductor Research Center, Dongguk University, Seoul 100-715, Republic of Korea; 2Post-Silicon Semiconductor Institute, Korea Institute of Science and Technology, Hwarang-ro 14 gil, Seoungbuk-ku, Seoul 130-650, Republic of Korea; 3Division of Physics and Semiconductor Science, Dongguk University, Seoul 100-715, Republic of Korea; 4Nanophotonics Research Center, Korea Institute of Science and Technology, Hwarangno 14-gil 5, Seongbuk-gu, Seoul 130-650, Republic of Korea; 5Department of Applied Physics, Kyung Hee University, Yong-In 130-701, Republic of Korea; 6Quantum-Function Research Laboratory and Department of Physics, Hanyang University, Seoul 133-791, Republic of Korea

**Keywords:** noble metal, InZnP:Ag nano-rods, ferromagnetism

## Abstract

InZnP:Ag nano-rods fabricated by the ion milling method were thermally annealed in the 250~350 °C temperature range and investigated the optimum thermal annealing conditions to further understand the mutual correlation between the optical properties and the microscopic magnetic properties. The formation of InZnP:Ag nano-rods was determined from transmission electron microscopy (TEM), total reflectivity and Raman scattering analyses. The downward shifts of peak position for LO and TO modes in the Raman spectrum are indicative of the production of Ag ion-induced strain during the annealing process of the InZnP:Ag nano-rod samples. The appearance of two emission peaks of both (A^0^ X) and (e, Ag) in the PL spectrum indicated that acceptor states by Ag diffusion are visible due to the effective incorporation of Ag-creating acceptor states. The binding energy between the acceptor and the exciton measured as a function of temperature was found to be 21.2 meV for the sample annealed at 300 °C. The noticeable MFM image contrast and the clear change in the MFM phase with the scanning distance indicate the formation of the ferromagnetic spin coupling interaction on the surface of InZnP:Ag nano-rods by Ag diffusion. This study suggests that the InZnP:Ag nano-rods should be a potential candidate for the application of spintronic devices.

## 1. Introduction

Dilute magnetic semiconductors (DMSs) are promising materials for spintronic applications (of course, QD, nanoRod, etc.) as the intrinsic spin of the electron is exploited in addition to the electron charge. The magnetic semiconductor contains a lot of compounds and GaMnAs is a good material to show ferromagnetic properties. However, the drawback is the low Curie temperature, of less than 200 K, which does not has a ferromagnetism at room temperature. Meanwhile, a new attempt to increase the transition temperature was carried out in this nano-rod material. The nano-rods were formed on the surface of InZnP:Ag after the diffusion of Ag. Over the past few decades, the ferromagnetic properties based on III-V materials doped with the transition metal (TM) and ferromagnetic elements were extensively investigated for obtaining ferromagnetism above room temperature (RT) [1,2,3,4,5,6,7]. According to the RKKY theory based on the Zener model, the strong interaction between electrons in the delocalized s- and p-type band, and hole and localized d-type electrons of magnetic ions, is called the s-, p-d exchange interaction. Mn-doped GaN and ZnO were reported to have a high Curie temperature (T_C_) above 300 K with a Mn concentration of 5% per cation in 2^+^ charge state and 3.5 × 10^20^ holes cm^−3^, and many experimental studies have also been actively carried out. The ferromagnetic transition, where T_C_ is approximately 50 K, has been interpreted as the RKKY interaction between Mn spins (S = 5/2), which is mediated by free holes, resulting in an estimation of the magnitude of parameter |N0ß|~3.3 eV [4], i.e., the sp-d exchange interaction.

Meanwhile, according to the other double exchange model, the total energy (TE) calculated within the local spin density approximation was calculated for both (Ga_1−x_TM_x_^↑^)N and (Ga_1−x_TM_x/2_^↑^ TM_x/2_^↓^)N as a function of x. For the compositions of x = 5, 10, 15, 20, and 25%, the ferromagnetic states of GaNMn, GaNV, and GaNCr were stable, but GaNCo, GaNNi, and GaNFe were not stable (spin-glass states). Particularly, the ferromagnetic states of GaNFe were not observed for all values of x. However, contrary to the reported theory, ferromagnetic properties in the GaNFe layer have been recently reported. Therefore, we selected the RKKY model rather than the double exchange model because the structure of InP and zinc-blende is the same as the structure of GaAs.

In view of the short history for Ag doping, the investment of InZnP:Ag nano-rod first is focused on experiments; in addition, there have been recently significant achievements in one dimensional (1-D) nanostructure semiconductors. These materials have interesting electrical and optical properties due to the unique structural one-dimensionality, where quantum confinement effects can occur. It has been reported that 1-D nanostructure semiconductors can be used in various nano-scale devices [8,9,10,11]. The size dependence can give rise to the magnetic properties of the material due to the various effects such as the influence of surface, the onset of carrier confinement, and the reduction in structure size below that of a single magnetic domain [12,13,14,15,16,17,18]. The magnetic properties of InP with 1D nanostructures doped by TM elements have been investigated for the application of nano-scale spintronics devices [19,20,21,22]. However, for applications in practical spintronic devices, a lot of research of InP based on DMS is needed. The nano-rod was easily produced by the ion milling method. The estimation on nanostructure for the magnetic properties of InZnP:Ag indicates that InZnP:Ag nano-rods are ferromagnetic at room temperature.

We have previously reported the formation of doped with noble metal Ag-doped InZnP nano-rods using the ion milling method, in which the Ag dopant was introduced to stable magnetic properties [23]. Transmission Electron Microscopy (TEM) for the structure and X-ray Photoelectron Spectroscopy (XPS) for electronic measurement of InZnP nano-rods doped with Ag revealed well-aligned nano-rods with a single crystalline zinc blend structure and a proper Ag incorporation into this nano-rod system. Furthermore, we found that Ag is substituted into In site by measuring triple-axis diffraction (TAD). None of the groups have so far explored the existence of ferromagnetism in Ag-doped InZnP nano-rods and their doping mechanism.

In this work, in order to understand the Ag doping mechanism and excitonic emission properties, the effect of thermal annealing on optical and ferromagnetic properties of InZnP:Ag nano-rods was investigated. Additionally, we explored the mutual correlation between the optical changes and the observed microscopic magnetic properties. 

## 2. Materials and Methods

The fabrication process of nano-rods was that the deposition with metal Ag was carried out using an e-beam evaporator, and then ion-milling of 20 nm was performed. By the Czochralski method, the p-type InP:Zn was grown, and then Ag was deposited onto the InP:Zn using an e-beam evaporator. After the deposition of Ag, thermal annealing had been performed in a flowing Argon (Ar) atmosphere, and the annealing temperature was varied from 250 to 350 °C for 30 min to investigate the effect of the annealing temperature.

For etching the entire surface of the sample, an ion miller, dedicated for wide-surface milling, was used. The initial pressure of the milling chamber was set to 3 × 10^−6^ Torr. Prior to milling the sample, the pre-milling without the sample was performed for 15 min to stabilize the miller. While the milling process was being conducted, the Ar gas pressure was set to 2 × 10^−4^ Torr. For preventing asymmetric etching, the sample stage was rotated, and the surface of the sample was etched out by using an ion miller made by Hitachi. The pre-milling process was conducted before the main milling for obtaining the stable milling rate. During the pre-milling, the samples were covered with shutter, and in the middle of the main milling, the argon gas flowed at a rate of 20 cc/min. For preventing the chemical damage on the surface, reactive gas was not used. The angle between the ion beam and the sample was tilted by 15°, and the sample holder continuously rotated to etch the surface evenly. A TEM measurement was performed in order to confirm the production of the nanostructure, namely, nano-rod. In addition, we investigated the total reflection of the InZnP:Ag sample to study the nano-rod structure. A Shimadzu UV-2600 system with an integrating sphere was used for total reflectance measurements. The spot size is 5 × 8 mm^2^. Temperature-dependent PL measurements were performed using a 0.92 W laser with a wavelength of 800 nm as the excitation source. The samples were mounted on a cold finger in a closed cycle He- cryostat to control the temperature between 20 and 300 K. The Raman signal for the Ag diffusion effect at the different thermal annealing temperature was measured by the micro-Raman system (Uni-nanotech). To investigate the correlation and agreement between the observed optical, exact changes, and the nano-scale magnetic domains of InZnP:Ag nano-rods, the local surface topography, and magnetic properties of InZnP, Ag nano-rods were analyzed by AFM and MFM, respectively. The AFM and MFM measurements were carried out at RT by the non-contact mode, and the Co/Cr coated tip was used and magnetized just before scanning. 

## 3. Results and Discussion

### 3.1. TEM and Reflectance Spectra

Figure 1a shows a schematic illustration of a fabrication process flow of the InZnP:Ag nano-rods. After the deposition of the Ag layer onto InZnP and subsequent annealing of the Ag layer, the Ag was partly inner-diffused and existed on InP:Zn. The Ag layer plays an important role in the formation process of InZnP:Ag nano-rods. It should be noted that the Ag layer on InP:Zn will protect the top shape of InZnP:Ag nano-rods to maintain the entire nano-rod formation under the ion-milling process [23]. After 20 nm ion milling, the entire formation of InZnP:Ag nano-rods with an average height 50 nm was confirmed using the cross-sectional TEM image, as shown in Figure 1b. The distinct and uniform nano-rods were very well-produced. Figure 1c presented the reflectance spectra of p-type InP:Zn(pristine) and InZnP:Ag nano-rod samples annealed at 250 (A), 300 (B), and 350 (C) °C. The samples of (A), (B), and (C) were reduced up to approximately 20%, and are related to the nano-rod structure formation compared with the p-type InP:Zn(Pristine). In general, it is well proven that the nano-rod structure acts as an anti-reflection effect and reduces their reflectivity. It can be also seen from Figure 1c that the absorption peaks are found near 2.3 eV in all nano-rod samples. These peaks are caused by the surface plasmon effect of residual Ag [24,25,26], which agrees with the results for the TEM image of Figure 1b—that InZnP:Ag nano-rods are well-formed.

### 3.2. PL Spectra

Figure 2a represents the PL spectra of the as-deposited Ag layer on InZnP (inset) and InZnP: Ag nano-rod samples annealed at 250, 300, and 350 °C at 20 K. For the as-deposited Ag layer on the InZnP sample, the dominant peak at 1.377 eV was observed, which is usually called the A1. The A1 peak corresponds to the unidentified donor carbon acceptor pair transition [27], and a broad emission peak around 0.96 eV is associated with the Ag-related peak [28]. However, for annealed InZnP:Ag nano-rod samples, a new peak around 1.398 eV related to acceptor-bound exciton (A^0^ X) appeared together with another emission peak around 1.377 eV. The emission peak around 1.337 eV belongs to the electron and to the neutral acceptor (e, Ag) transition. Similar peaks corresponding to Mn [27,29] acceptors in InP were previously reported in similar positions. After the samples were annealed at 250, 300, and 350 °C, the PL peak displayed suppression of Ag-related peaks around 0.96 eV compared with (A^0^ X) and (e, Ag). Therefore, the two emission peaks of both (A^0^ X) and (e, Ag) were newly produced due to Ag dopants introduced by Ag diffusion, which also means an improvement in crystallinity because of the effect of heat treatment. It is indicated that acceptor states by Ag diffusion are visible due to the effective incorporation of Ag-creating acceptor states, which are further confirmed by the observation of hole-induced ferromagnetism in our previous work [23]. It can be explained that the ferromagnetic property of InZnP:Ag is generated by the substitution of the Ag element into In cation sites. The substitution of Ag into In made free hole, and the spin coupling of ferromagnetic property is caused by the mediation by free hole, namely:(1)$$Ag_{In}^{+3}\;\left(4d^{10}\right)=A^{0}\to Ag_{In}^{+2}\;\left(4d^{9}\right)+\mathrm{hole}\to A^{-}+\mathrm{hole}$$

Figure 2b depicted or demonstrated the peak position of (A^0^ X), A1, and (e, Ag) as a function of the annealing temperature of the InZnP:Ag nano-rods measured at 20 K. This specific plot allows to quantify the variation for a peak position of (A^0^ X), A1, and (e, Ag). The peak positions of (A^0^ X), A1, and (e, Ag) in the sample that annealed at 300 °C were observed to be blue shift compared to the annealed at 250 °C samples. On the other hand, in the sample annealed at 350 °C, the peak positions of (A^0^ X), A1, and (e, Ag) were observed to be red shift compared to the annealed at 300 °C sample. This change in peak position indicates that the lattice change in InZnP:Ag nano-rods is caused due to the Ag ion diffusion after the annealing process. Moreover, these results mean that the effective Ag ion diffusion took place well in the sample that annealed at 300 °C.

In order to further understand the assignment of PL emission and acceptor binding energy of InZnP:Ag nano-rods according to the annealing temperature, a temperature-dependent PL measurement over the 20 to 290 K was carried out for all samples. The results are displayed in Figure 2c for the annealed 300 °C samples. The peak intensity of the (A^0^ X), A1, and (e Ag) decreased with substantial peak broadening and demonstrated a red shift as the temperature increased to above 140 K. It is expected that this is primarily due to the exciton–phonon interaction and partially due to thermal expansion [30,31]. With the increasing temperature above 160 K, both (A^0^ X) and A1 gradually decreased and shifted to red, and peaks existed at RT to lower-emission energy positions of 1.330 eV and 1.337 eV, respectively. The presence of (A^0^ X) at RT indicates that the formation of a stable acceptor level is due to Ag doping introduced by Ag diffusion. A similar trend is evident for the sample annealed at 250 °C and 350 °C. The appearance of the (e, Ag) emission enables us to calculate the accepter binding energy (E_bA_) at 20 K using,
E_bA_ = E_gap_ – E_eA_ + k_B_T/2,(2)
where E_eA_ is the temperature-dependent transition, and E_eA_
*=* 1.335 eV, 1.337 eV, and 1.336 eV at 20 K for samples annealed at 250, 300, and 350 °C, respectively, with an intrinsic bandgap of E_gap_ = 1.423 eV; the value of E_bA_ is calculated to be 19.3 meV, 21.2 meV, and 22.3 meV for 250, 300, and 350 °C annealed samples, respectively. These values are almost comparable to those of Mn-doped InP [27,29].

### 3.3. Raman Spectra

In addition to the compositional changes on the lattice of InZnP by Ag diffusion, we can also conjecture crystal structural changes induced by Ag due to defects, disorders, and even lattice deformation. Raman spectroscopy is sensitive to crystal structure changes. Additionally, it can provide useful information of nanostructure material. The confinement of first-order optical phonon wave functions results in the lowering of phonon frequency and asymmetric line-shape broadening. Based on the first-order Raman spectrum of III-V semiconductors such as InP and other zinc-blende structures, we have been able to investigate the evolution of the optical phonon mode and nano-rod formation as a function of annealing temperature. Figure 3 represented the Raman spectra and the shift of LO and TO mode in pristine InP:Zn (100) and InZnP:Ag nano-rod samples annealed at 250, 300, and 350 °C, respectively. In Figure 3a, the Raman spectrum of pristine InP:Zn (100) displayed only one strong longitudinal optical (LO) phonon mode at 346.9 cm^−1^. The transverse optical (TO) phonon mode (around 300 cm^−1^) faintly appeared in the pristine InP (100) face. However, for the InZnP:Ag nano-rod samples annealed at 250, 300, and 350 °C, a new peak around 300 cm^−1^ related to the TO phonon mode was observed along with the LO mode shifts downward by 1.2, 6 and 3.6 cm^−1^, respectively, compared to the pristine InP sample. The appearance of the TO mode in InZnP:Ag samples reflects the formation of the nano-rods due to a breakdown of polarization selection rules [32]. At the same time, the LO mode shifts indicate that the (lnZnP:Ag nano-rods) lattice is under stress due to the Ag ion diffusion after the annealing process. The largest LO mode peak shift of 6 cm^−1^ was observed for the InZnP:Ag nano-rod sample annealed at 300 °C. This means that the proper Ag ion diffusion occurred in the sample annealed at 300 °C (and the surface is under large tensile stress). This result is consistent with the change in peak position for (A^0^ X), A1, and (e, Ag) in PL. Additionally, the TO mode in the InZnP:Ag nano-rod samples annealed at 300 and 350 °C, respectively, were shifted toward lower wavenumbers by 1.3 cm^−1^ compared to the annealed at 250 °C sample, as shown in Figure 3b. Moreover, the asymmetric Raman shapes of LO and TO phonon modes indicate the presence of nanostructure and disordered nanostructure. In addition, the LO and TO phonon wave function confinements in nanostructure and surface nano-rods can lower the phonon frequencies. The downward shift of a peak position of LO and TO modes are indicative of the development of Ag ion-induced strain during the annealing process of the InZnP:Ag nano-rod samples. Similar trends have also been reported for InP quantum dots [33].

### 3.4. AFM and MFM

For investigating the effect of the magnetic coupling of Ag atoms and the related surface of nanoscale magnetic domains in the InZnP:Ag nano-rods, the AFM and MFM measurements were carried out under RT for all (annealed) samples. In view of the PL and Raman results for the InZnP:Ag nano-rods, the AFM and MFM measurements were representative of the sample annealed at 300 °C because the samples annealed at different temperatures revealed a similar behavior in comparison with the sample annealed at 300 °C. Typical AFM and MFM images of the InZnP:Ag nano-rods annealed at 300 °C are presented in Figure 4. The clearly distinct individual nano-rod features are seen in the AFM 3D image, as shown in Figure 4a. Based on the AFM results, the average height of InZnP:Ag nano-rods, including residual Ag on the top surface, was approximately 50 nm (ag20, rod20). This result coincides with the cross-sectional TEM result in Figure 1b. Figure 4b showed the MFM 3D image for the InZnP:Ag nano-rods. The remarkable contrast in the MFM image implies the surface magnetic activity produced by Ag diffusion in InZnP:Ag nano-rods. At the sub-micrometer scale, the contrast of MFM images is caused by the force gradients between the MFM tip and the magnetic activity on the surface of the sample. In this study, the MFM images were obtained after topography measurements (non-contact mode) followed by InZnP:Ag nano-rod scanning at a constant 50 nm height (lift mode). In course of the process, van der Waals forces are not expected to be detected, and the change in the amplitude fude of vibration of the cantilever is proportional to the gradient of magnetic fields perpendicular to the sample surface [34]. Moreover, the ion milling sample of InP:Zn without the Ag layer and Ag as-deposited onto InP:Zn sample were not observed in the MFM image contrast under the same measurement conditions. It is indicated that the observed surface nanoscale magnetic activity of InZnP:Ag nano-rods originates from the formation of the spin coupling interaction of ferromagnetism in InP:Zn doped with Ag by diffusion. The magnetism is not generated by defect-induced magnetism due to ion milling and the secondary phase, such as InAg, AgP, etc., of which are generated by diamagnetism [35]. Therefore, it is confirmed that the image contrast shown in Figure 2b is obvious evidence of the surface nanoscale magnetic activity present in InZnP:Ag nano-rods. The MFM image (Figure 4b) indicates the top side of the nano-rods as a response, according to the MFM tip. Thus, the aligning was carried out in one direction in the surface. Figure 4c displayed the AFM height profile and MFM phase obtained under diagonal scans along the dashed line drawn in Figure 4a,b. It is interesting to observe quite different AFM topographic and MFM phase patterns that have arisen from the same scanning line due to the presence of magnetic activity. Such a clear distinction between the AFM topographic and MFM phase patterns observed, according to the scanning distance, is attributed to the presence of magnetic activity formed on the surface of InZnP:Ag nano-rods by Ag diffusion. The AFM (MFM) image indicates a qualitative analysis, but the AFM(MFM) data need a quantitative analysis in more information using frequency domain filtering and the least squares method [36].

The ferromagnetic types of Mn centers which can be possibly formed in III-V compounds were found in some bulk GaAs:Mn and InP:Mn samples [37,38]. The type of mechanism, i.e., $Mn_{Ga}^{+3}\;\left(3d^{4}\right)+hv\;\to Mn_{Ga}^{2+}\;\left(3d^{5}\right)+hole=A^{0}+hv\;\to \;A^{-}+hole$ is suggested. It can be explained that the ferromagnetic property of InZnP:Ag can be formed by the substitution of Ag into the In cation site. The substitution of Ag into In produces free hole, and ferromagnetic spin coupling results from the mediation by free hole, namely, $Ag_{In}^{+3}\;\left(4d^{10}\right)$ = A^0^ → $Ag_{In}^{+2}\;\left(4d^{9}\right)$ + hole → A^−^ + hole, based on the RKKY model rather than the double exchange model at present.

## 4. Conclusions

InZnP:Ag nano-rods were fabricated by the ion milling method, and the nano-rods were thermally annealed in temperature ranging from 250 to 350 °C. The structural, optical, and magnetic properties of the nano-rods were investigated by TEM, Raman scattering, total reflectivity, PL, AFM, and MFM. For the annealed nano-rod samples, the produced two emission peaks of both (A^0^ X) at 1.398 eV and (e, Ag) at 1.336 eV may take place due to Ag dopants introduced by Ag diffusion. It was indicated that acceptor states by Ag diffusion can be formed due to the effective incorporation of Ag. The value of binding energy between the acceptor and the exciton was 21.2 meV for the sample annealed at 300 °C. The downward shifts of peak position for LO and TO modes led to the production of Ag ion-induced strain during the annealing process of the nano-rod samples. The noticeable MFM image contrast and the clear change in MFM phase with the scanning distance indicated the formation of the ferromagnetic spin coupling interaction on the surface of the nano-rods by Ag diffusion. The analysis of these experimental results revealed that Ag was effectively incorporated into the nano-rods and that a thermal annealing temperature of 300 °C was the optimum temperature for Ag diffusion. The origin of the ferromagnetic property of the nano-rods can be related to the p-d spin coupling caused by free hole according to Ag substitution into In site. This study suggested that nano-rods should be a potential candidate for the application of spintronic devices. 

## Figures and Tables

**Figure 1 nanomaterials-12-04200-f001:**
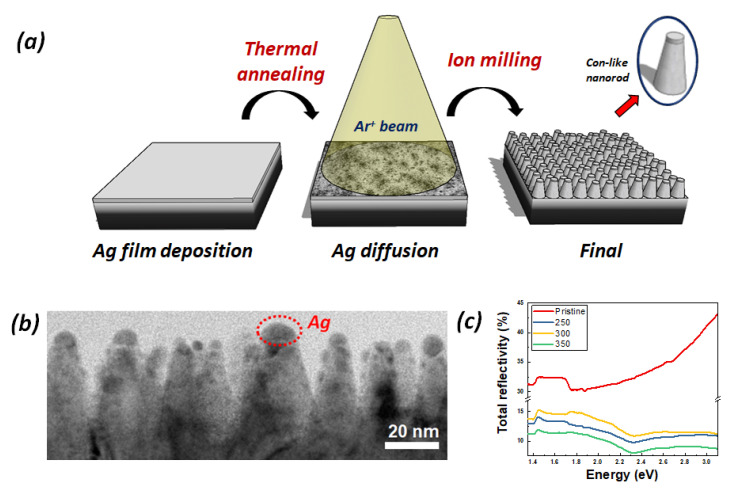
(**a**) Schematics illustration of the InZnP:Ag nano-rod fabrication process. (**b**) The cross-sectional TEM image of InZnP:Ag nano-rods after 20nm ion milling. (**c**) The total reflectivity of InP:Zn(pristine) and InZnP:Ag nano-rods annealed at 250, 300, and 350 °C.

**Figure 2 nanomaterials-12-04200-f002:**
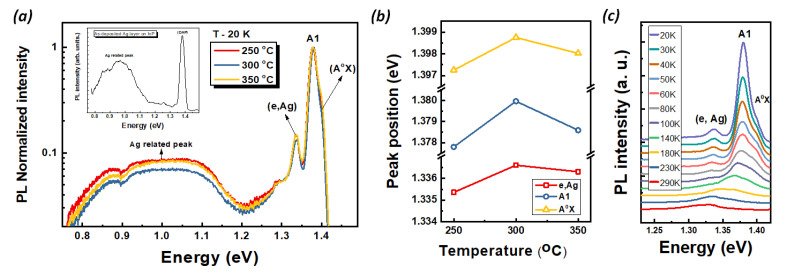
(**a**) PL spectra of the as-deposited Ag layer on InZnP(inset) and InZnP:Ag nano-rod samples annealed at 250, 300, and 350 °C measured at 20 K. (**b**) The peak position of (A^0^ X), A1, and (e, Ag) as a function of the annealing temperature of the InZnP:Ag nano-rods measured at 20 K. (**c**) The temperature-dependent PL spectra of InZnP:Ag nano-rod samples annealed at 300 °C.

**Figure 3 nanomaterials-12-04200-f003:**
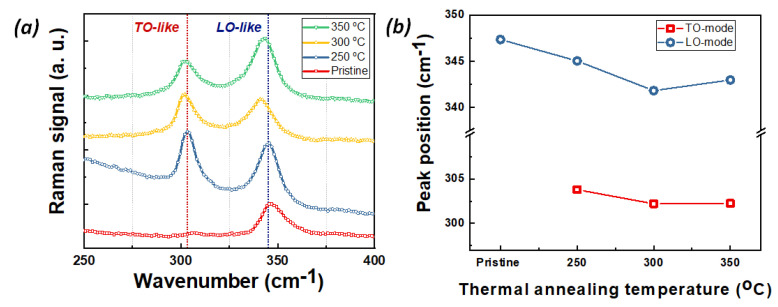
(**a**) Raman spectra and (**b**) the peak position of LO and TO mode in pristine InP:Zn (100) and InZnP:Ag nano-rod samples annealed at 250, 300, and 350 °C, respectively.

**Figure 4 nanomaterials-12-04200-f004:**
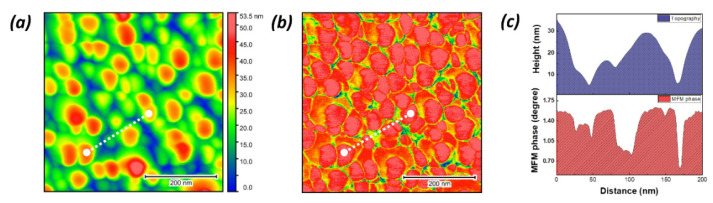
(**a**) AFM and (**b**) MFM images of the InZnP:Ag nano-rods annealed at 300 °C. (**c**) AFM height profile and MFM phase obtained under diagonal scans along the dashed line drawn in (**a**) and (**b**).

## Data Availability

Not applicable.

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
