# Peer review of "Effects of Thermal Annealing on Optical and Microscopic Ferromagnetic Properties in InZnP:Ag Nano-Rods"

_nanomaterials, 2022, doi:10.3390/nano12234200_

Round 1

Reviewer 1 Report

In this work, you studied effect of thermal annealing of InZnP:Ag nanorods in the temperature range of 250~350℃ on the mutual correlation between optical and microscopic magnetic properties. It was interesting to read your Manuscript.

In my opinion, to improve the quality of the Manuscript, it is desirable to answer the following questions:

1. What is the novelty of your work?

2. In References, you list only one work about InZnP:Ag. Therefore:

2.1) why did this material attract your attention?

2.2) do the obtained results prove the advantages of this material compared to other materials, which can be applied in spintronic devices?

Author Response

Dear Editor,

Thank you for your decision letter on our manuscript, Manuscript ID: nanomaterials-2011118, Title:“Effects of thermal annealing on optical and microscopic ferromagnetic properties in InZnP:Ag nano-rods”. Owing to your helpful comments, our manuscript could be corrected and more improved.

Introduction section was described in more details according to reviewer’s comment, and the response for the comments of MFM was addressed in the manuscript. The revised section was marked in red color. English expressions of the manuscript were improved with a help of editing from the service company.

We hope that the revised manuscript is suitable for publication in Nanomaterials.

Thank you very much for your considerations.

Sincerely yours,

                                                 Prof. Yoon Shon

Quantum-functional Semiconductor Research Center, Dongguk University

1.What is the novelty of your work?

Answer) The nano-rods were formed on the surface of InZnP:Ag after diffusion of Ag. The nano-rod was easily produced by ion milling method. The estimation on nano-structure for the magnetic properties of InZnP:Ag indicates that InZnP:Ag nano-rods is ferromagnetic at room temperature.

2.1) Why did this material attract your attention?

Answer) The research area of magnetic semiconductor contains a lot of compounds including GaMnAs. However, the Curie point for the operation of magnetic devices should be room temperature. As a try, research of a nano-structure can be performed to identify the ferromagnetic properties at room temperature. So, this work is attractive for replacement of earlier reported materials.

2.2) Do the obtained results prove the advantages of this material compared to other materials, which can be applied in spintronic devices?

Answer) The magnetic semiconductor contains a lot of compounds, GaMnAs is a good material to show ferromagnetic properties. But the drawback is the low Curie temperature, less than 200K, not to function at room temperature. Meanwhile, new attempt to increase the transition temperature was done in this nano-rod materials. So, this work is applicable for the room temperature operating devices.

Reviewer 2 Report

The paper presents interesting optical and microscopic ferromagnetic properties in InZnP:Ag nano-rods. I think the current work is almost ok for publication.

A minor question to ask: the magnetic domain in the MFM image (Fig 4b) seems to be all aligning in one direction, which is quite unusual (i.e. magnetic domain always up and down to minimize the magnetic energy without magnetic field), any reason for that? Maybe the author could discuss that in the paper too!

Author Response

Dear Editor,

Thank you for your decision letter on our manuscript, Manuscript ID: nanomaterials-2011118, Title:“Effects of thermal annealing on optical and microscopic ferromagnetic properties in InZnP:Ag nano-rods”. Owing to your helpful comments, our manuscript could be corrected and more improved.

Introduction section was described in more details according to reviewer’s comment, and the response for the comments of MFM was addressed in the manuscript. The revised section was marked in red color. English expressions of the manuscript were improved with a help of editing from the service company.

We hope that the revised manuscript is suitable for publication in Nanomaterials.

Thank you very much for your considerations.

Sincerely yours,

                                                 Prof. Yoon Shon

Quantum-functional Semiconductor Research Center, Dongguk University

- The paper presents interesting optical and microscopic ferromagnetic properties in InZnP:Ag nano rods. I think the current work is almost ok for publication.

A minor question to ask: the magnetic domain in the MFM image (Fig4b) seems to be all aligning in one direction, which is quite unusual(i.e. magnetic domain always up and down to minimize the magnetic energy without magnetic field), any reason for that? Maybe the author could discuss that in the paper too.

Answer) Thank you for your valuable comments for our manuscript. We have revised the manuscript according to your comments and added more detail description into the manuscript. The MFM image (Fig 4b) indicates the top side of the nano-rods as a response according to MFM tip. So the aligning was done in one direction in the surface. We added the explanation to contents.